# CAD-Coder: Text-to-CAD Generation with Chain-of-Thought and Geometric Reward

**Yandong Guan**
School of Software
Beihang University
Beijing, China
yd_guan@buaa.edu.cn

**Xilin Wang**
School of Software
Beihang University
Beijing, China
wang_xilin@buaa.edu.cn

**Ximing Xing**
School of Software
Beihang University
Beijing, China
ximingxing@buaa.edu.cn

**Jing Zhang**
School of Software
Beihang University
Beijing, China
zhang_jing@buaa.edu.cn

**Dong Xu**
The University of Hong Kong
Hong Kong, China
dongxu@cs.hku.hk

**Qian Yu**[*]
School of Software
Beihang University
Beijing, China
qianyu@buaa.edu.cn

## Abstract

In this work, we introduce CAD-Coder, a novel framework that reformulates text-to-CAD as the generation of CadQuery scripts—a Python-based, parametric CAD language. This representation enables direct geometric validation, a richer modeling vocabulary, and seamless integration with existing LLMs. To further enhance code validity and geometric fidelity, we propose a two-stage learning pipeline: (1) supervised fine-tuning on paired text–CadQuery data, and (2) reinforcement learning with Group Reward Policy Optimization (GRPO), guided by a CAD-specific reward comprising both a geometric reward (Chamfer Distance) and a format reward. We also introduce a chain-of-thought (CoT) planning process to improve model reasoning, and construct a large-scale, high-quality dataset of 110K text–CadQuery–3D model triplets and 1.5K CoT samples via an automated pipeline. Extensive experiments demonstrate that CAD-Coder enables LLMs to generate diverse, valid, and complex CAD models directly from natural language, advancing the state of the art of text-to-CAD generation and geometric reasoning.

## 1 Introduction

Computer-Aided Design (CAD) systems are fundamental tools in engineering and manufacturing, enabling the creation of precise 3D models. However, traditional CAD workflows often demand significant expertise and are time-consuming [21, 6], which limits broader accessibility and hampers rapid iteration. Recent advancements in Large Language Models (LLMs)[28], particularly their proficiency in natural language understanding and code generation[3, 1], present a promising opportunity to streamline CAD processes [5, 26] based on natural language descriptions. The ability to generate or modify CAD models via textual instructions [2, 13, 16] could lower the entry barrier for novices and enhance the efficiency of experienced users.

However, generating CAD from textual descriptions remains a nontrivial challenge. To leverage progress in natural language processing, researchers have proposed representing CAD models using pre-defined command sequences and formulating text-to-CAD as a machine translation problem—i.e., autoregressively predicting CAD command tokens conditioned on input text [32, 13].

---

[*]Corresponding author.

39th Conference on Neural Information Processing Systems (NeurIPS 2025).

Despite their utility, these approaches face several limitations. First, verifying the validity of a CAD model represented by command sequences is challenging. Second, most existing methods support only a limited set of operations, such as *sketch* and *extrusion*, restricting the diversity of generated CAD models. Third, CAD command sequences are often difficult to interpret and edit, complicating both understanding and debugging.

To address these issues, we advocate for a new proxy representation of CAD models. In this paper, we utilize CadQuery [7], a Python-based parametric CAD scripting language, as the target representation. CadQuery is particularly suitable for the following reasons: (1) It provides inherent geometric validation, as CadQuery scripts can be directly executed to verify the validity of the resulting CAD model. (2) It offers a rich vocabulary for CAD modeling, enabling the representation of diverse and complex geometries. (3) CadQuery scripts are composed of semantic, function-based constructs, making them more interpretable than low-level command sequences. (4) Importantly, as CadQuery is implemented in Python, it allows us to leverage the code generation capabilities of modern LLMs that are already proficient in programming tasks.

Consequently, we reformulate the text-to-CAD task as generating CadQuery code from natural language input. While this representation enables the use of LLMs for CAD generation, adapting LLMs to reliably produce high-quality CAD models remains challenging. The core difficulty arises from the dual requirements of CadQuery code: syntactic correctness (from a programming perspective) and geometric plausibility (from a 3D modeling perspective). While supervised fine-tuning (SFT) on paired text and CadQuery code can teach the model syntactic patterns, it is insufficient to guarantee both code validity and geometric correctness, as it lacks explicit 3D knowledge and reasoning capabilities [39, 9, 38].

To overcome these challenges, we draw inspiration from recent advances where reinforcement learning (RL) has improved LLM reasoning and planning across various domains. In particular, we integrate Group Reward Policy Optimization (GRPO), an efficient RL algorithm, into the text-to-CAD code generation pipeline. Our approach consists of two stages: (1) We begin by supervised fine-tuning an LLM with paired natural language descriptions and CadQuery code to establish basic syntax and mapping. (2) We then enhance the model's planning and reasoning ability via RL, introducing a novel CAD-Specific reward function.

Specifically, since multiple distinct CadQuery scripts can produce geometrically equivalent CAD models—a challenge for SFT to capture—we introduce a chain-of-thought (CoT) process that encourages the model to plan before code generation. Our CAD-Specific reward comprises two components: a *geometric* reward, which uses the Chamfer Distance (CD) between generated and target 3D geometries to ensure geometric accuracy, and a *format* reward, which enforces the reasoning process and syntactic correctness of the generated code.

To facilitate research in this area, we construct a large-scale, geometrically verified dataset comprising 110K *text–CadQuery-3D model* triplets and 1.5K high-quality CoT samples. We also propose an automatic data construction pipeline to accelerate dataset creation and ensure high quality. Extensive experiments demonstrate that our method unlocks new capabilities for LLMs, enabling the generation of complex, functional CAD models directly from high-level textual intent. In summary, our contributions include the following:

- We propose a novel approach **CAD-Coder** that reformulates the text-to-CAD task as generating CadQuery code from natural language descriptions. Leveraging the Python-based CadQuery enables more interpretable, diverse, and valid CAD model generation, while fully utilizing the code generation capabilities of existing large language models.

- We introduce a two-stage pipeline that combines supervised fine-tuning with reinforcement learning using GRPO. Our method incorporates a chain-of-thought (CoT) planning process and a novel CAD-Specific reward, which jointly enforce both syntactic correctness and geometric plausibility in the generated CAD models.

- We construct a high-quality, large-scale dataset consisting of 110K verified text–CadQuery-3D model triplets and 1.5K CoT samples via an automated pipeline, facilitating further research in text-to-CAD generation and geometric reasoning.

## 2 Related Work

### 2.1 Large Language Model for Code Generation

Large language models (LLMs) have revolutionized code generation, with models like GPT-4 [20] and specialized code-focused LLMs such as CodeLlama [27] showcasing impressive capabilities in translating natural language into various programming languages [3, 1]. Standard training typically involves supervised fine-tuning (SFT) on extensive code corpora and instruction datasets. To better align LLM behavior with specific goals or complex tasks, reinforcement learning (RL) techniques have been increasingly employed. Reinforcement learning with human feedback (RLHF)[11] is widely used for general alignment. For more task-specific optimization, policy gradient algorithms like Proximal Policy Optimization (PPO)[23] are commonly utilized, though these often require training a separate critic network, which adds computational overhead.

Our approach leverages Group Reward Policy Optimization (GRPO) [24], a more recent and efficient RL algorithm that estimates baselines through relative rewards within a sample batch, removing the need for a critic and making RL fine-tuning more feasible for complex tasks like ours. Generating structured and logically coherent code, especially for multi-step procedures common in CAD modeling, requires advanced reasoning. Chain-of-Thought (CoT) prompting [30] has proven effective in improving the reasoning and planning capabilities of LLMs by prompting them to generate intermediate steps. We leverage CoT to enhance the model's ability to decompose complex natural language instructions into coherent CadQuery code sequences.

### 2.2 CAD Generation

Generative modeling for CAD systems commonly employs two main representations: boundary representation (B-rep) [14] and command sequence representations [32, 31]. B-rep models combine geometry and topology to offer high precision and accuracy; however, their complexity presents significant challenges for generative modeling. To address this, various models use separate latent spaces and decoders for geometry and topology [10, 36, 8]. Notably, HoLa [17] introduces a unified latent space for B-rep generation. Despite their advantages, direct generation of B-rep models from text remains computationally intensive, as it involves capturing intricate geometric features and interrelationships. Alternatively, command sequence representations, such as those proposed by DeepCAD [32], model the procedural nature of CAD design by encoding the design process as a series of commands, e.g., sketch creation or extrusion. Several approaches [12, 4] have demonstrated the ability to generate command sequences from point clouds or images. CAD-MLLM [35] leverages multimodal large language model (MLLM) to enhance the performance of this generation process. In the context of text-to-CAD generation, methods like Text2CAD [13] and CAD-Translator [16] use encoder-decoder architectures to translate textual descriptions into command sequences. Additionally, CAD-Llama [15] and CADFusion [29] employ LLMs to further address the complexity of this task. Recent efforts also study controllability in CAD generation: FlexCAD [41] unifies control across construction hierarchies via structured-text serialization and LLM fine-tuning, while GeoCAD [40] targets local geometry controllability with a mask-and-predict paradigm over captioned local parts. While command sequence representations simplify CAD model generation, they often lack direct connections to the geometric accuracy of the resulting models.

In contrast, our approach leverages CadQuery [7], a Python-based parametric library that facilitates programmatic CAD model generation. This representation offers significant advantages, including enhanced editability, better interpretability, and compatibility with LLMs. CAD-Recode [22] generates CadQuery scripts from point clouds, while Query2CAD [2] directly prompts LLMs to produce CadQuery code from text. CAD-Assistant [19], instead, adopts a tool-augmented VLLM setup built on the FreeCAD Python API to execute modeling operations. These works follow different technical routes; we focus on CadQuery as an intermediate code representation for text-to-CAD. Beyond 3D CAD, vector graphics (SVG) constitute a closely related parametric and programmatic design space. Recent studies on SVG understanding and generation leverage executable code representations and learning signals, e.g., LLM4SVG [34] and Reason-SVG [33], underscoring the value of interpretable, code-like geometry—an idea that resonates with adopting CadQuery in our setting.

# 3 Methodology

## 3.1 CadQuery: CAD Representation as Python Code

We adopt CadQuery, a Python-based parametric CAD scripting language, as the core representation for 3D modeling in our framework. CadQuery can be executed directly without any external software dependencies. CadQuery allows models to be constructed using chainable geometric operations (*e.g.*, `box()`, `circle()`, `extrude()`), encoded as modular and readable Python code. Each script corresponds to a complete, executable modeling procedure that can be rendered directly via the OpenCascade kernel into high-fidelity 3D geometry.

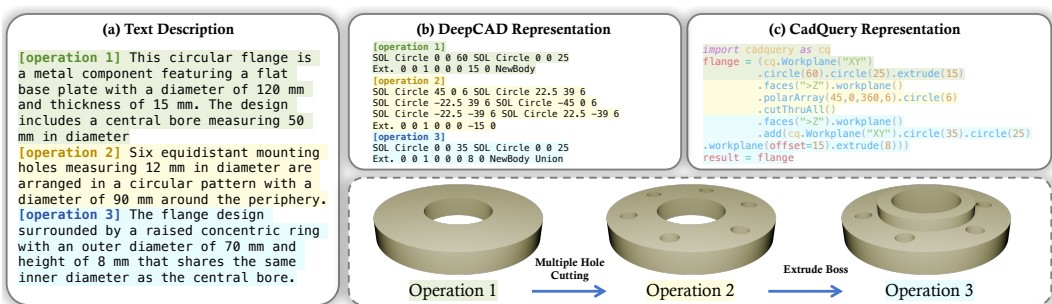

Figure 1: (a) Text description of a CAD model. (b) Corresponding *sketch-extrusion* command sequence used in DeepCAD. (c) Corresponding CadQuery code used in our method. The bottom row shows the resulting 3D models generated by each of the three sequential operations.

Traditional methods such as DeepCAD [32] represent model structures using sketch-extrude command sequences. As illustrated in Fig. 1, these representations are typically linearized and low-level, lack modularity, and cannot be directly executed. They often require additional post-processing to produce 3D shapes. This not only increases modeling complexity but also hinders the model's ability to learn structured modeling semantics. In contrast, CadQuery is interpretable and executable. CadQuery provides expressive and flexible geometric operations, ranging from basic primitives to complex modeling procedures. Meanwhile, CadQuery scripts can be directly executed for validation. The provided high-level API can also better align with the input textual description.

Considering the Python nature, CadQuery is well-suited for generative modeling with language models. Nevertheless, a crucial challenge lies in generating code that is both syntactically correct and produces geometrically accurate and valid 3D designs. To address this, our approach employs a two-stage training strategy. First, an initial Supervised fine-tuning (SFT) phase teaches the model the specific CadQuery syntax. Then, a reinforcement learning (RL) phase is introduced to further enhance the geometric accuracy and validity of the generated 3D output.

## 3.2 CAD-Coder

**Overview.** We adopt Qwen2.5-7B-Instruct [37], a strong open-source language model pre-trained on a mixture of web text and code, as the base model for CadQuery code generation. The input to the model is a natural language description $L$ of the 3D design intent. The output is an executable CadQuery script $C$ which is executed to produce a 3D geometry $M = \text{Execute}(C)$. The model is fine-tuned and optimized in an autoregressive decoding setup, where CadQuery tokens are generated sequentially, conditioned on the input and previous tokens.

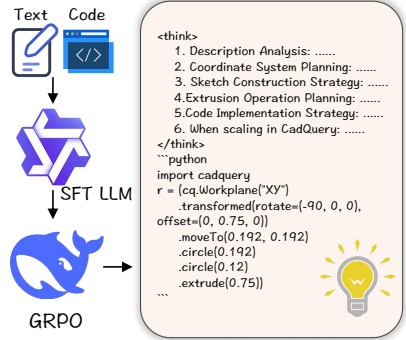

Figure 2: CAD-Coder Training Pipeline

To better generate CAD models, our method leverages a two-stage training strategy. In the first stage, we SFT the LLM with paired data, which enables the model to learn CadQuery's fundamental syntax and common programming patterns. To further enhance the

geometric reasoning ability of the model, which improves the accuracy of the final 3D model, we introduce the second RL stage with a reward specifically designed for CAD.

**Stage 1: Supervised Fine-Tuning for CAD Code Generation.** We begin by performing SFT to equip the model with the basic capability to translate natural language descriptions into executable CadQuery code. Unlike generic code generation, CAD code must follow strict syntactic and geometric constraints. This phase serves as a foundation that enables the model to understand the CadQuery's syntax and learn the basic mapping between high-level descriptions to low-level modeling operations in a structured format.

We train on a synthetic high-quality dataset containing 8k examples generated through our data annotation pipeline (see Section 4). Each training sample is a pair $(L, C_{gt})$, where $L$ is a natural language prompt and $C_{gt}$ is the corresponding ground-truth CadQuery code that has been verified for executability and filtered by geometric correctness.

The model learns to generate a predicted CAD program $C = \{c_t\}_{t=1}^{|C_{gt}|}$ token-by-token, aiming to approximate the groundtruth program $C_{gt}$ as closely as possible. We adopt a standard autoregressive decoding framework, where the model learns to predict each token of $C_{gt}$ sequentially given the prompt $L$ and the preceding tokens. Formally, the loss function is:

$$\mathcal{L}_{\text{SFT}}(\theta) = -\mathbb{E}_{(L,C_{gt})\sim\mathcal{D}_{\text{SFT}}} \left[ \sum_{t=1}^{|C_{gt}|} \log \pi_\theta(c_t \mid c_{<t}, L) \right] \tag{1}$$

where $L$ denotes the input prompt, $C_{gt} = \{c_t\}_{t=1}^{|C_{gt}|}$ represents the ground-truth code sequence, $c_{<t}$ refers to the preceding tokens before step $t$, $\pi_\theta$ denotes the model policy with parameters $\theta$, and $|C_{gt}|$ is the length of the code sequence.

This allows the model to follow the syntax of CadQuery and establish preliminary mappings between common shape-related language patterns and CAD primitives (*e.g.* "create a hole" $\rightarrow$ `.hole()`, "draw a circle" $\rightarrow$ `.circle()`).

After this stage, the model shows promising capabilities in generating valid CadQuery code for standard and relatively simple modeling cases. However, we observe two major limitations: The generated code sometimes lacks geometric accuracy compared to the target shape, and the model struggles with complex structures that require multi-step or spatial reasoning.

**Stage 2: Reinforcement Learning with CAD-Specific Rewards.** To address these challenges, we introduce a CAD-Specific RL stage using Group Reward Policy Optimization (GRPO) to improve the geometric reasoning capability, enhanced with chain-of-thought (CoT) prompting to guide structured reasoning. We first cold-starting the model with designed CoT samples to enhance basic reasoning ability. Then, during RL phase, we specifically introduce Chamfer Distance(CD), a common geometric metric, into the reward signal to directly optimize the model based on the 3D output quality instead of token-level loss.

**CoT Design.** Unlike direct prompt-to-code pairs used in standard SFT, CoT samples are formatted as $(L_{\text{cot}}, C)$, where $L_{\text{cot}}$ includes a step-wise plan embedded in natural language before the final modeling instruction. Considering the hierarchical and compositional nature of CAD modeling, we design $L_{\text{cot}}$ to simulate an engineer's planning process, including component decomposition, coordinate system assignment, sketch design, and extrusion operations each outlined succinctly within <think>...</think> tags. This structured reasoning format helps the model map textual descriptions to executable geometry. These intermediate reasoning steps guide the LLM to break down complex shapes into simpler components aligned with textual description.

**Reward Design.** During GRPO training, for each input $L_{\text{cot}}$, the current policy $\pi_\theta$ generates $k$ diverse CadQuery candidates $\{C_1, \dots, C_k\}$. The final CAD-Specific reward $R_i$ includes two components: geometric reward $R_i^{\text{geo}}$, and format reward $R_i^{\text{fmt}}$.

In contrast to program synthesis tasks where GRPO uses exact match-based rewards, CAD modeling lacks a unique ground-truth code. Multiple solutions can yield identical geometry. To address this, we design a geometric reward based on CD between the rendered 3D model and the target geometry $M_{gt}$. For each generated code candidate $C_i$, we first attempt execution using the CadQuery engine. If successful, the resulting mesh $M_i$ is uniformly sampled into a dense point cloud. Similarly, $M_{gt}$ is

also sampled. The CD is then computed between these two point clouds as:

$$\text{CD}(P, Q) = \frac{1}{|P|} \sum_{x \in P} \min_{y \in Q} \|x - y\|_2^2 + \frac{1}{|Q|} \sum_{y \in Q} \min_{x \in P} \|x - y\|_2^2 \tag{2}$$

where $P$ and $x$ are sampled from the predicted shape, and $Q$ and $y$ from the ground-truth shape; $|P|$ and $|Q|$ denote the number of points in each set.

This metric quantifies the geometric discrepancy between the generated shape and the ground-truth model. Smaller CD values indicate closer geometric alignment.

To convert CD values into reward signals, we define a piecewise geometric reward $R_i^{\text{geo}}$. Specifically, if the CD is smaller than $1 \times 10^{-5}$, the candidate is assigned the maximum reward of 1.0. When the CD is greater than to 0.5, or if the code fails to execute, the reward is set to 0. For CD values between these thresholds, the reward decreases linearly: a CD of 0.5 corresponds to a minimum non-zero reward of 0.01, and smaller CD values yield proportionally higher rewards. This design ensures that the model receives continuous geometric feedback, encouraging approximate yet geometrically close solutions even when exact reconstruction is difficult.

To compute format reward $R_i^{\text{fmt}}$, we apply regular expression matching to detect whether the output $C_i$ contains both a `<think>...</think>` reasoning block and a properly formatted Python code block (delimited by triple backticks ```python ... ```). If both are present, $R_i^{\text{fmt}} = 1$; otherwise, $R_i^{\text{fmt}} = 0$.

To compute final reward, each CadQuery candidate $C_i$ is executed, valid outputs are converted into 3D models, and CD is computed with respect to $M_{gt}$. The combined reward $R_i = \lambda_{\text{geo}} R_i^{\text{geo}} + \lambda_{\text{fmt}} R_i^{\text{fmt}}$ is used to compute the relative advantage, and the model is updated via the GRPO loss (Eq. 3):

$$\mathcal{L}_{\text{GRPO}}(\theta) = \mathbb{E}_{L_{\text{cot}} \sim \mathcal{D}, \ \{C_i\}_{i=1}^k \sim \pi_{\theta_{\text{old}}}(\cdot | L_{\text{cot}})}$$

$$\left[ \frac{1}{k} \sum_{i=1}^k \frac{1}{|C_i|} \sum_{t=1}^{|C_i|} \min\left( r_{i,t}(\theta) \cdot \hat{A}_{i,t}, \ \text{clip}(r_{i,t}(\theta), \ 1 - \varepsilon, \ 1 + \varepsilon) \cdot \hat{A}_{i,t} \right) - \beta D_{\text{KL}}(\pi_\theta \parallel \pi_{\text{ref}}) \right] \tag{3}$$

where $L_{\text{cot}}$ is the input prompt, $C_i$ is a sampled code sequence, $k$ is the number of samples per prompt, $t$ is the token index, $\theta$ is the current policy parameter, $\theta_{\text{old}}$ is the old policy parameter, $\hat{A}_{i,t}$ is the advantage estimate, $\varepsilon$ is the clipping threshold, $\beta$ is the KL penalty weight, $\pi_\theta$, $\pi_{\theta_{\text{old}}}$, and $\pi_{\text{ref}}$ are the current, old, and reference policies.

This training strategy allows the model to continuously refine its code generation towards executable, semantically meaningful, and geometrically accurate CAD outputs.

## 4 Dataset Construction

We build our dataset based on the Text2CAD dataset [13], which contains 178K natural language descriptions $L$ paired with ground-truth 3D geometries $M_{gt}$. However, Text2CAD lacks executable CadQuery code aligned with $M_{gt}$, which poses a major obstacle for training models that generate script-based CAD representations.

To address this, we design a CadQuery data annotation pipeline, as shown in Fig. 4. For each sample, we take the CAD command sequence $S$ provided by Text2CAD (which is structurally aligned with $M_{gt}$) and prompt a code-generation LLM (DeepSeek-V3 [24]) to generate multiple candidate CadQuery scripts. We attempt to execute each candidate using the CadQuery engine and discard failures. The successfully executed models $M_{\text{cand}}$ are compared against $M_{gt}$ using CD, and we select the candidate with the lowest CD as the final $C_{\text{gt}}$. Note that the mapping from a command sequence (or $M_{gt}$) to CadQuery code is one-to-many: multiple syntactically different scripts can produce geometrically equivalent shapes. Our selection by minimum CD explicitly preserves *geometric equivalence* while allowing script diversity; thus $C_{\text{gt}}$ is a verified executable surrogate aligned to $M_{gt}$ rather than a unique textual ground truth.

In total, this pipeline produces 110K valid triplets $(L, C_{\text{gt}}, M_{gt})$. We further divide them into three subsets based on geometric quality: 8k high-quality samples with $\text{CD}_{\text{gt}} < 1 \times 10^{-4}$; 70k medium-quality samples with $\text{CD}_{\text{gt}} < 1 \times 10^{-3}$; and the remaining 32k hard cases with $\text{CD}_{\text{gt}} > 1 \times 10^{-3}$.

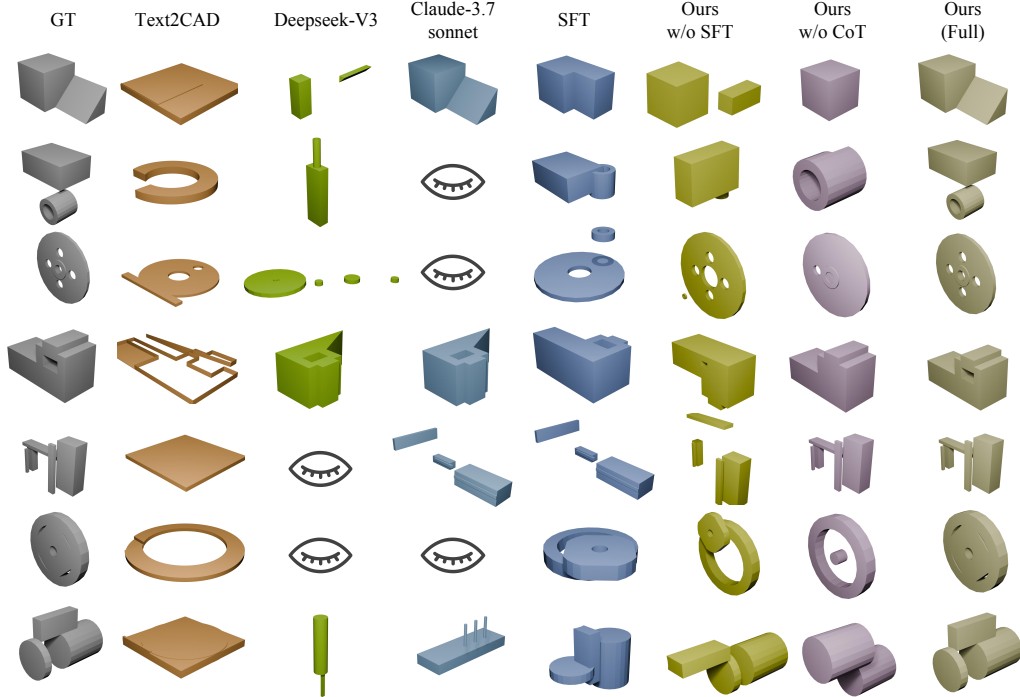

| GT | Text2CAD | Deepseek-V3 | Claude-3.7 sonnet | SFT | Ours w/o SFT | Ours w/o CoT | Ours (Full) |

Figure 3: Qualitative comparison between baseline methods and different model variants under various training strategies.Text2CAD is a command-sequence-based baseline; Deepseek-V3 and Claude-3.7 represent open-source and proprietary LLMs, respectively. The right columns show our ablations and ull model, which best preserve structure and geometry.

To bootstrap reinforcement learning and enhance structured reasoning, we further construct a set of CoT-formatted samples on complex shapes. Specifically, we select the hard cases from the dataset, use their description $L$ to prompt DeepSeek-V3 for CoT-style CadQuery code. We leverage the same filtering strategy as Fig. 4, retaining samples that are executable and exhibit high geometric accuracy based on CD. Each valid candidate is further manually refined for correctness. The final CoT dataset contains 1.5K high-quality CoT samples.

## 5   Experiments

**Metrics.**    For the Text-to-CAD task, we use metrics adapted from prior 3D generation works [13]: (1) **Mean CD** measures the average geometric discrepancy between the generated and ground-truth models over sampled point clouds.  (2) **Median CD** captures the typical geometric error and is more robust to outliers.  (3) **Invalidity Ratio (IR)** denotes the proportion of generated CadQuery programs that fail to be executed to yield valid 3D geometry. These metrics jointly capture geometric fidelity and executable correctness. We evaluate on the official Text2CAD test split. Let $L$ denote the dataset-provided text prompt. Given $L$, our model generates a CAD program $C_{\text{pred}}$ and its mesh $M_{\text{pred}}$. Following [13], we apply the same normalization and compute CD between $M_{\text{pred}}$ and the ground-truth mesh $M_{\text{gt}}$. Importantly, our translated CadQuery scripts are never used as test ground truth.

**Implementation Details.**    We use Qwen2.5-7B-Instruct [37] as the base model for all experiments, given its strong instruction-following and code generation capabilities. For the stage of SFT, we fine-tuned Qwen2.5-7B-Instruct for 3 epochs with a batch size of 64 and a learning rate of $1 \times 10^{-5}$, using the AdamW optimizer [18]. Training was performed using full-parameter fine-tuning with DeepSpeed ZeRO Stage 2. For the GRPO phase, we initialized the model with SFT weights and trained for 1 epoch with a batch size of 384. To enable cold-starting of reasoning during SFT, we additionally fine-tuned the model on the 1.5K high-quality CoT-format samples for 2 epochs.

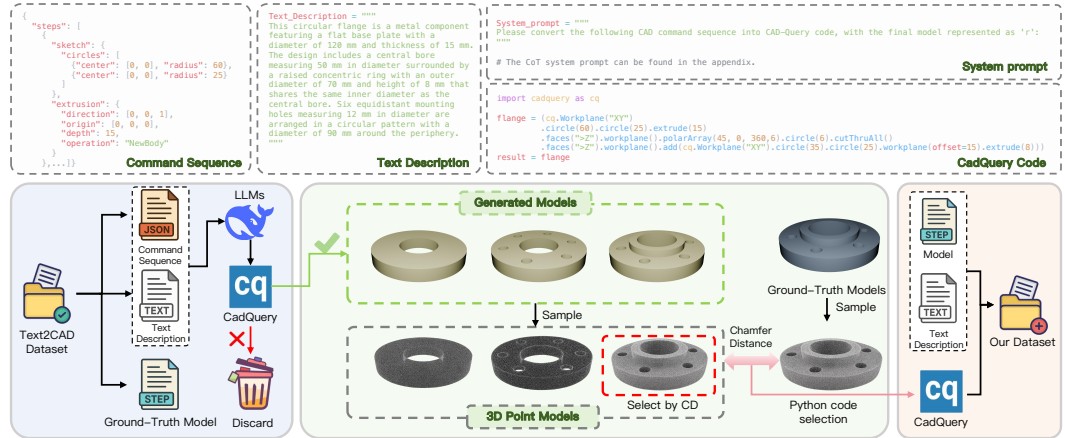

Figure 4: Overview of our annotation pipeline. Given CAD command sequences and natural language descriptions from the Text2CAD dataset, we use DeepSeek-V3 to synthesize multiple CadQuery code candidates. These candidates are executed and compared to the ground-truth 3D models using the Chamfer Distance (CD). Scripts that execute successfully and achieve the lowest CD are retained. Finally, we construct a dataset comprising text–CadQuery–3D model triplets.

The batch size was set to 384. Each input prompt generated $k = 8$ candidate completions. The KL divergence coefficient was set to $\beta = 0.001$. All geometric computations, including model execution via CadQuery [7], point cloud sampling, normalization, and CD calculation, following the Text2CAD [13] implementation to ensure consistency. We utilized the Hugging Face Transformers library, GRPO implementation from Verl [25], and DeepSpeed for distributed training.

# 6 Extended Ablation Studies

For SFT, we use the 8K high-quality samples. For cold-starting, we use the 1.5K CoT-format samples. For GRPO, we use all 150K training descriptions and geometries from Text2CAD. For evaluation, we apply the same synthesis pipeline on the official Text2CAD test set to obtain corresponding triplets.

**Baselines.** We compare our full method (SFT+CoT+GRPO) against several baseline methods for text-to-CAD generation. Text2CAD [13] directly generates CAD models from natural language descriptions. We also evaluate several LLMs by prompting them directly with natural language descriptions to generate CadQuery code, without any fine-tuning. The baseline LLMs include open-source models Qwen2.5-72B, Qwen2.5-7B, DeepSeek-V3, as well as the proprietary models Claude-3.7-sonnet, GPT-4o. For parity, all LLM baselines are prompted with the *same* CoT-style format used by CAD-Coder; the full prompt is provided in Appendix C.

Table 1: Quantitative comparison on the test set. CD metrics are $\times 10^3$. IR.% indicates Code Invalidity Ratio. Lower CD and lower IR.% are better.

| Method | Mean CD ↓ | Median CD ↓ | IR.% ↓ |
|---|---|---|---|
| Claude-3.7-sonnet | 186.53 | 134.16 | 47.03 |
| GPT-4o | 143.5 | 40.96 | 70.5 |
| Deepseek-V3 | 186.69 | 107.57 | 51.96 |
| Qwen2.5-72B | 209.41 | 153.81 | 82.64 |
| Qwen2.5-7B | 202.35 | 169.86 | 98.83 |
| Text2CAD [13] | 29.29 | 0.37 | 3.75 |
| CAD-Coder (Ours) | **6.54** | **0.17** | **1.45** |

## 6.1 Main Results

Table 1 summarizes the quantitative performance on the test set. Our full method achieves the best results across all metrics, significantly outperforming prior works in terms of geometric accuracy. Specifically, it reduces the Mean CD to 6.54 and the Median CD to 0.17, both by large margins compared to the strong baseline Text2CAD [13], surpassing all existing LLMs. Fig. 3 exhibits the qualitative results. We can observe that LLMs frequently fail to generate valid code. Our method can better align with the target shape. These results highlight the effectiveness of our geometry-aware optimization and CoT-enhanced reasoning in generating precise and structurally valid 3D CAD models. Our method also maintains a lower code invalidity ratio, demonstrating that reinforcement-driven learning does not compromise executability. All experiments were conducted on 8 NVIDIA A800 80GB GPUs. Additional hyperparameter details are provided in the Appendix. With vLLM-based serving and KV caching, average decoding latency remains below $1\,\mathrm{s}$ on mainstream GPUs (H800, A800, RTX 4090, V100) for both CoT-enabled and SFT decoding. The reported timings reflect token generation only and exclude code execution. Per-GPU results are provided in Table 2.

Table 2: Per-sample inference latency (seconds; lower is better). *CoT* denotes CoT-enabled decoding; *SFT* denotes plain decoding without CoT.

| GPU Model | CoT (s) ↓ | SFT (s) ↓ |
|---|---|---|
| H800 80G | 0.06 | 0.03 |
| A800 80G | 0.18 | 0.12 |
| RTX 4090 24G | 0.28 | 0.16 |
| V100 32G | 0.64 | 0.29 |

## 6.2 Ablation Study

Table 3 isolates the effect of each component in the training pipeline. Starting from a model trained solely with SFT, we observe limited geometric fidelity (Mean CD 74.55, Median CD 0.33). This baseline demonstrates that SFT alone cannot adequately capture spatial reasoning for complex CAD structures. However, the model also performs poorly (Mean CD 76.20) without SFT, showing the necessity of using prior knowledge. Even without CoT, the GRPO alone dramatically boosts performance (Mean CD 17.34, Median CD 0.20), confirming that CAD-Specific reward supervision is essential for improving 3D accuracy. Our full method, adding CoT prompting for cold-starting, further improves results (Mean CD 6.54, Median CD 0.17), indicating that structured multi-step reasoning enhances the model's ability to handle complex geometric prompts. Fig. 5 illustrates the CD distributions of the generated model under different training strategies. We can observe that more effective training strategies result in distributions that are skewed towards smaller CD values and fewer invalid results. Fig. 3 reveals the visualization results with different components. Experimental results demonstrate the effectiveness of each component.

Table 3: Ablation study results on the test set using Qwen2.5-7B-Instruct. CD metrics are $\times 10^3$.

| Training Strategy | Mean CD↓ | Med CD↓ | IR %↓ |
|---|---|---|---|
| SFT | 74.55 | 0.33 | 5.33 |
| Ours w/o SFT | 76.20 | 0.95 | 5.33 |
| Ours w/o CoT | 17.34 | 0.20 | 4.95 |
| Ours (Full) | 6.54 | 0.17 | 1.45 |

Table 4: Ablation study: effect of sft training data quality.

| Dataset | Mean CD↓ | Med CD↓ | IR %↓ |
|---|---|---|---|
| Ours w/ 70K | 9.89 | 0.18 | 3.21 |
| Ours w/ 8K | 6.54 | 0.17 | 1.45 |

In Table 4, we explore how different SFT training data affect final model performance. Training with the full 70K medium-quality dataset during SFT leads to substantial improvement over existing methods (Mean CD 9.89). However, training with a smaller but high-quality 8K dataset yields the best result (Mean CD 6.54, Median CD 0.17), outperforming the larger dataset. These results reveal a key insight: quality outweighs quantity. High-precision data offers better foundation for CAD-Specific RL, considering that small inconsistencies in code can lead to significant errors in CAD geometry.

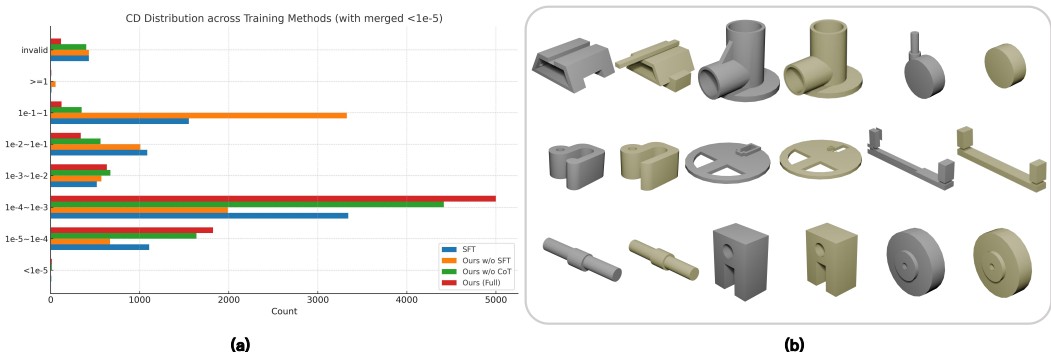

(a)                                                 (b)

Figure 5: (a) Chamfer Distance (CD) distributions of generated CAD models trained with different strategies. (b) Visualizations of predicted CAD models across three CD intervals. Gray shapes represent ground-truth models, while brown shapes denote generated models. The **first row** shows results with $\mathrm{CD} > 1 \times 10^{-1}$, indicating that the generated CAD models differ substantially from the ground truth. The **second row** presents models with $1 \times 10^{-4} < \mathrm{CD} \leq \times 10^{-1}$, and the **third row** displays models with $\mathrm{CD} \leq 1 \times 10^{-4}$, indicating that these models are nearly identical to the ground-truth models.

## 7   Conclusion

In this paper, we have presented a novel approach to text-to-CAD generation by leveraging CadQuery as an intermediate representation. By combining the strengths of Python-based code generation and the inherent interpretability of CadQuery, our method overcomes key challenges associated with traditional command sequence-based approaches, including model validity and limited operation sets. We propose a two-stage training strategy combining supervised fine-tuning (SFT) with reinforcement learning (RL). The integration of Group Reward Policy Optimization (GRPO) and a CAD-Specific reward function ensures that the generated CAD models are both syntactically correct and geometrically plausible, while the Chain-of-Thought (CoT) process allows for improved reasoning and planning. Our large-scale, geometrically verified dataset facilitates further research in this domain, and the experimental results show that our method significantly advances the capabilities of LLMs in generating complex CAD models from natural language descriptions. This work opens the door to more accessible, efficient, and flexible CAD generation, making it easier for both novice and experienced users to create high-quality 3D models based on textual input.

**Limitations and future work:** While CAD-Coder achieves strong performance, several limitations remain. First, the model currently does not support multimodal inputs such as images or point clouds, which restricts its applicability in real-world design scenarios. Second, although CoT prompting enhances reasoning, it remains shallow and often fails on extremely complex spatial compositions. Third, the reward design primarily relies on Chamfer Distance and format checks, limiting fine-grained structural supervision. Future work could extend the framework to a broader set of CAD operations and further optimize LLM planning and reasoning.

## Acknowledgments

This work was supported in part by the National Key Research and Development Project of China (No. 2022ZD0117801) and the Young Elite Scientists Sponsorship Program by CAST, in part by the National Natural Science Foundation of China (Nos. 62572039, 62461160331, and 62132001), and in part by the Fundamental Research Funds for the Central Universities. This work was also supported by the NSFC/RGC Collaborative Research Scheme (CRS_HKU703/24). Dr. Xu's research work described in this paper was conducted in the JC STEM Lab of Multimedia and Machine Learning, funded by The Hong Kong Jockey Club Charities Trust.

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

## Overview of Supplementary Material

This supplementary material provides additional details in support of our main paper, *CAD-Coder: Text-to-CAD Generation with Chain-of-Thought and Geometric Reward*. The contents are organized as follows:

- In Section A, we describe the hardware and software configurations, training durations for different stages, and the Chamfer Distance (CD) evaluation protocol.

- In Section B, we show that using only Chamfer Distance as reward leads to training failure, emphasizing the importance of multi-faceted reward design.

- In Section C, we present a detailed breakdown of the CAD generation process, including the user prompt and CoT prompt, CoT reasoning steps, and the final CadQuery output.

- In Section D, we provide additional qualitative comparisons between different methods.

- In Section E, we present several examples to show that our CAD-Coder supports CAD editing.

- In Section F, we analyze failure cases where our method underperforms.

## A    Additional Implementation Details

All experiments were executed on a cluster equipped with 8 NVIDIA A800 (80GB) GPUs. The SFT stage was trained for 7 hours, while the GRPO stage required 146 hours, both utilizing distributed training with standard data parallelism techniques, facilitated by DeepSpeed and Ray. For efficient model inference, we employed vLLM, and used CadQuery (version 2.3.1) for CAD script execution and validation.

Chamfer Distance (CD) was computed using the same implementation as in Text2CAD [13], ensuring a fair and reproducible comparison. The CD calculation relies on the normalization of the generated 3D mesh models, which is critical for consistent metric evaluation. While CAD-Translator [16] and CAD-LLaMA [15] address similar tasks, their implementations are not open-sourced, and the details of their normalization procedures remain unclear. As a result, although their papers report CD scores, the values differ by an order of magnitude compared to those reported by Text2CAD, making direct comparison infeasible.

## B    Extended Ablation Studies

To further analyze the impact of our proposed reward design, we conducted an ablation study by disabling all auxiliary components and retaining only the Chamfer Distance (CD) as the reward function during the GRPO training phase. All other settings, including model architecture, optimization strategy, and data pipeline, remained unchanged.

However, as training progressed, we observed that the model began to generate invalid CadQuery code after approximately 200 steps. This led to frequent failures during the reward evaluation stage, as the CD computation relies on successful execution of the generated scripts to produce mesh outputs. Once invalid syntax or structural inconsistencies occurred, the reward pipeline could no longer provide feedback, which ultimately caused the reinforcement learning to halt prematurely.

These results highlight the critical role of robust code supervision and reward shaping beyond pure geometry-based metrics, especially in code-generation scenarios where executability directly impacts learning stability.

# C  Step-by-Step CoT Reasoning in CAD Generation

To illustrate the step-by-step reasoning performed by our model prior to generating CadQuery scripts, we present an example corresponding to the first CAD model shown in Fig. 3. The user-provided prompt specifies precise geometric operations, which our model parses into a structured CoT (Chain-of-Thought) reasoning sequence. This sequence includes part decomposition, coordinate planning, sketch construction, and extrusion logic. The CoT reasoning is followed by executable CadQuery scripts that constructs the final 3D model. This example demonstrates the model's ability to convert complex, multi-step textual descriptions into valid, parameterized CAD scripts through interpretable and systematic reasoning.

---

### Input

**User Prompt:** *"Start by creating a new coordinate system for the first part, setting the Euler angles to [0.0, 0.0, -90.0] and the translation vector to [0.0, 0.0, 0.0]. Next, draw a two-dimensional sketch on a new face, forming a loop with four lines: the first line starts at (0.0, 0.0) and ends at (0.375, 0.0); the second line starts at (0.375, 0.0) and ends at (0.375, 0.375); the third line starts at (0.375, 0.375) and ends at (0.0, 0.375); and the fourth line starts at (0.0, 0.375) and ends at (0.0, 0.0). Apply a sketch scale of 0.375 to this loop. Rotate the scaled sketch using the Euler angles [0.0, 0.0, -90.0] and translate it using the translation vector [0.0, 0.0, 0.0]. Finally, extrude the sketch 0.375 units in the direction opposite to the normal to create a solid body with final dimensions of 0.375 units in length, width, and height. For the second part, create another new coordinate system, setting the Euler angles to [-90.0, 0.0, -90.0] and the translation vector to [0.375, 0.125, 0.0]. Draw a new two-dimensional sketch on a new face, forming a loop with three lines: the first line starts at (0.0, 0.0) and ends at (0.25, 0.0); the second line starts at (0.25, 0.0) and ends at (0.25, 0.25); and the third line starts at (0.25, 0.25) and ends at (0.0, 0.0). Apply a sketch scale of 0.25 to this loop. Rotate the scaled sketch using the Euler angles [-90.0, 0.0, -90.0] and translate it using the translation vector [0.375, 0.125, 0.0]. Extrude the sketch 0.375 units in the direction of the normal and add this extruded shape to the existing cube. The final dimensions of this rectangular prism are 0.25 units in length, 0.25 units in width, and 0.375 units in height."*

**CoT Prompt:** *Please create a CadQuery Python code to generate a model based on the following description. The reasoning process MUST BE enclosed within <think> </think> tags. The final CadQuery code MUST BE put in "'python code "' with ONLY the executable code inside the python box, nothing else. The final model is represented by r. In the <think> section, simulate the thought process of an engineer converting text descriptions into a CAD model. Follow these steps:*

➤ **Stage 1: Description Analysis.** Break down the description into different parts or components. Identify key parameters for each part (coordinate systems, Euler angles, translation vectors). Understand the spatial relationships and assembly sequence between parts.

➤ **Stage 2: Coordinate System Planning.** Determine the coordinate systems used for each part. Parse how Euler angle rotations and translation vectors are applied. Ensure understanding of local-to-global coordinate system transformations.

➤ **Stage 3: Sketch Construction Strategy.** Analyze how to create each 2D sketch (loops, lines, points). Determine scaling factors for each sketch. Plan how to transform sketches into 3D space.

➤ **Stage 4: Extrusion Operation Planning.** Identify extrusion parameters for each part (direction, distance). Understand how to add extrusions to existing solids (new or merge). Verify dimensions after extrusion match the description.

➤ **Stage 5: Code Implementation Strategy.** Plan the sequence of CadQuery operations. Determine necessary CadQuery functions and methods. Consider how to organize code for clarity and readability.

---

> ➤ **Stage 6: When scaling in CadQuery.** Directly scale the size, define a scaling factor variable, and apply the scaling factor directly to all coordinates and dimensions. Don't try to use a non-existent '.scale()' method on Workplane objects.
>
> *After your thinking process, provide clean, working CadQuery Python code to create the 3D model.Think step by step, but only keep a minimum draft for each thinking step, with 50 words at most. description:*

## Output

**CoT Reasoning Steps (Generated by CAD-Coder):**

```
1. **Description Analysis**:
   - Two parts: first rectangular prism, second triangular
     prism
   - First part has dimensions 0.375*0.375*0.375, rotated -90
     about Z
   - Second part is 0.25*0.25*0.375, rotated -90 about X and Z
     , translated to (0.375,0.125,0)
2. **Coordinate System Planning**:
   - First part uses identity CS, second rotated -90 X/-90 Z
     at (0.375,0.125,0)
   - Both parts translate to origin initially
3. **Sketch Construction Strategy**:
   - First part: 4-line rectangle sketch
   - Second part: 3-line triangular sketch
   - Apply scaling factors (0.375 and 0.25) to coordinates
4. **Extrusion Operation Planning**:
   - First part extruded -0.375Z
   - Second part extruded +0.375Z
   - Combine using union operation
5. **Code Implementation Strategy**:
   - Create base sketches
   - Apply rotations and translations
   - Extrude and combine
```

**Generated CadQuery Script:**

```python
import cadquery as cq
# First part
part1 = (cq.Workplane("XY")
        .moveTo(0.0, 0.0)
        .lineTo(0.375, 0.0)
        .lineTo(0.375, 0.375)
        .lineTo(0.0, 0.375)
        .close()
        .extrude(-0.375)
        .rotate((0,0,0), (1,0,0), 90)
        .translate((0.0, 0.0, 0.0))
    )
# Second part
part2 = (cq.Workplane("XY")
        .moveTo(0.0, 0.0)
        .lineTo(0.25, 0.0)
        .lineTo(0.25, 0.25)
        .close()
        .extrude(0.375)
        .rotate((0,0,0), (1,0,0), 90)
        .rotate((0,0,0), (0,0,1), 90)
        .translate((0.375, 0.125, 0.0))
    )
# Combine parts
r = part1.union(part2)
```

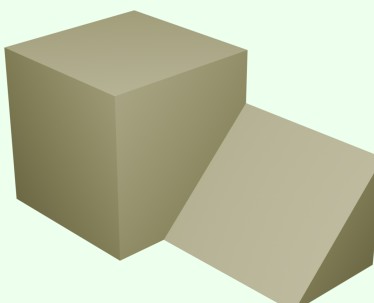

Figure S1: Generated CAD Model

# D  Additional Qualitative Comparisons Across Methods

Fig. S2 presents additional qualitative comparisons of CAD models generated by different methods. For methods that do not produce valid 3D models, placeholders are shown in their respective results.

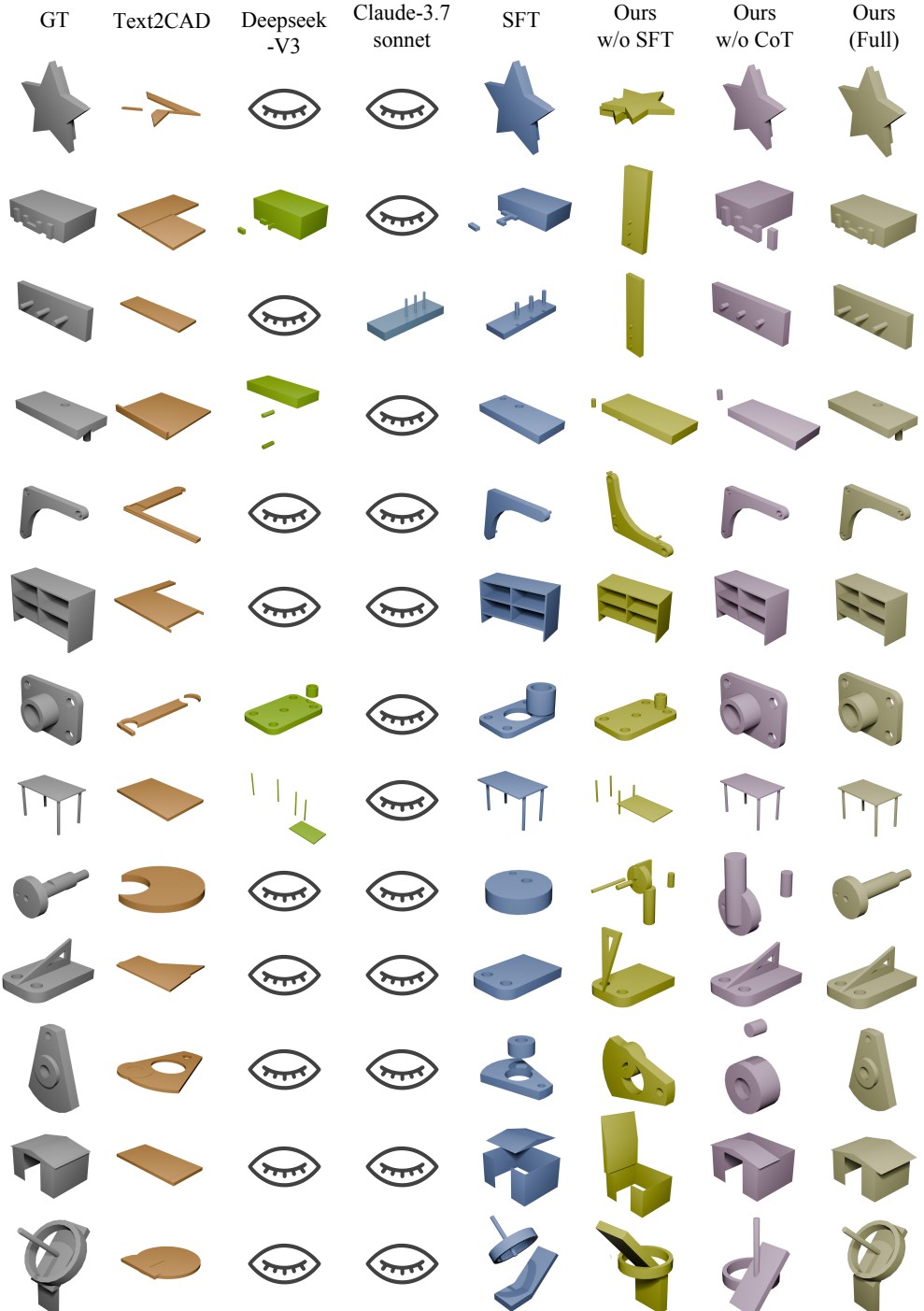

Figure S2: Additional qualitative comparisons of CAD models generated by different methods.

# E Performance on CAD Editing

Although our model was not explicitly trained on CAD editing data, it demonstrates promising capabilities in handling simple CAD editing tasks based on user instructions. This suggests that the model has acquired a degree of structural understanding of CadQuery code and can generalize beyond the training objective of generation-from-scratch.

As shown in Fig. S3, the model successfully performs lightweight operations such as modifying object dimensions, removing a component, or adjusting translation and rotation parameters in response to natural language prompts. These preliminary results highlight the model's potential to be extended toward interactive or instruction-following CAD editing scenarios.

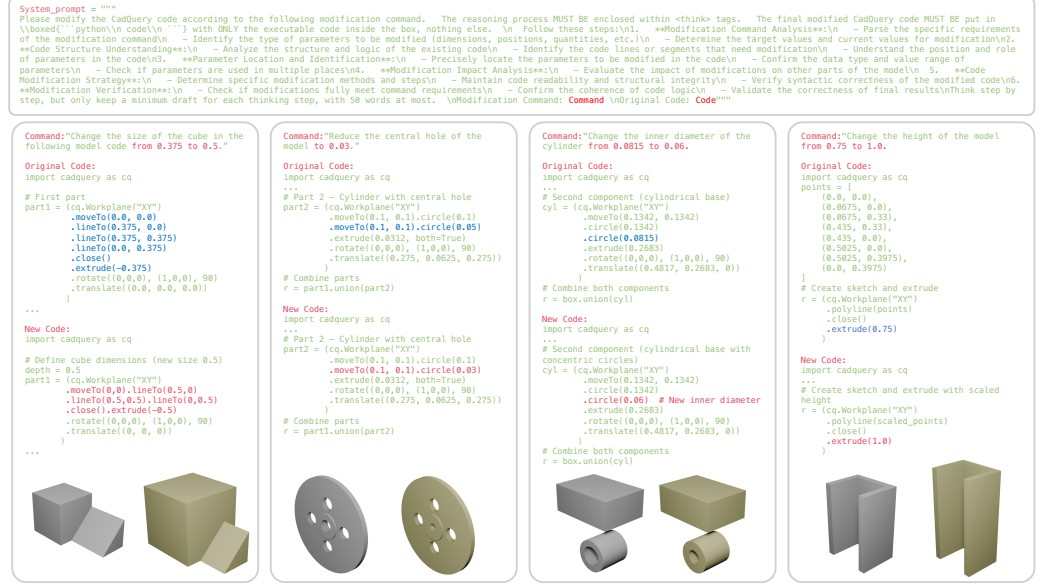

Figure S3: Examples of simple CadQuery code editing based on instructions.

# F Failure Cases

As shown in Figure S4, our method still struggles with certain challenging cases. As illustrated in Fig. S4(a), our method still struggles with complex structures composed of multiple sub-components, where inaccurate spatial alignment between modules can lead to visible dislocations or offsets. Moreover, as shown in Fig. S4(b), the model may misclassify operations such as extrusion and cutting, resulting in geometries that deviate from the intended design. In addition, Fig. S4(c) highlights the challenge posed by very thin structures or internal cavities, where sparse point sampling may induce reward hacking behavior, revealing limitations in handling overlapping features and tight geometric tolerances.

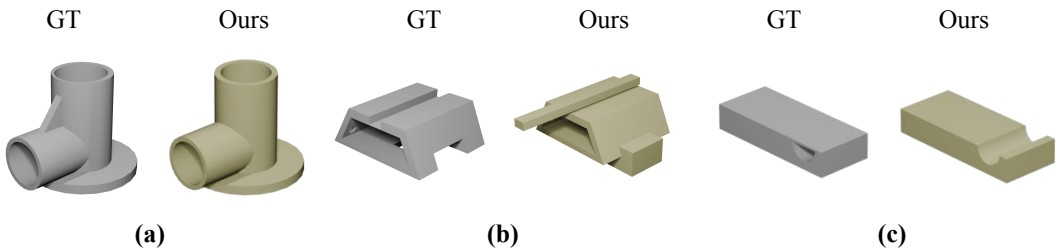

Figure S4: Examples of failure cases.

