# OpenReview forum: "CAD-Coder: Text-to-CAD Generation with Chain-of-Thought and Geometric Reward"
_NeurIPS.cc/2025/Conference — NeurIPS 2025 poster_

### Official Review · Reviewer_75fC · 2025-06-29

**Clarity:** 3
**Significance:** 2
**Originality:** 3
**Rating:** 4
**Confidence:** 3

**Summary:**

In this work, the authors introduce CAD-Coder, a novel framework that reformulates text-to-CAD as the generation of CadQuery scripts—a Python-based, parametric CAD language. This representation enables direct geometric validation, a richer modeling vocabulary, and seamless integration with existing LLMs. To further enhance code validity and geometric fidelity, the authors propose a two-stage learning pipeline: (1) supervised fine-tuning on paired text–CadQuery data, and (2) reinforcement learning with Group Reward Policy Optimization (GRPO), guided by a CAD-specific reward comprising both a geometric reward (Chamfer Distance) and a format reward. They also introduce a chain-of-thought (CoT) planning process to improve model reasoning, and construct a large-scale, high-quality dataset of 110K text–CadQuery–3D model triplets and 1.5K CoT samples via an automated pipeline. Extensive experiments demonstrate that CAD-Coder enables LLMs to generate diverse, valid, and complex CAD models directly from natural language, advancing the state of the art of text-to-CAD generation and geometric reasoning.

**Questions:**

1. How about the failure cases of this method? What is the common failure case of this method looks like?

2. How long does it take to generated a single mesh?

3. How long does it take to train GRPO?

**Ethical Concerns:**

["NO or VERY MINOR ethics concerns only"]

**Limitations:**

1. Currently, the generated shapes are rather simple, compared to SOTA 3D generation results like *Trellis: Structured 3D Latents for Scalable and Versatile 3D Generation*.

2. The method only supports text-to-CAD, instead of sketch-to-CAD.

**Paper Formatting Concerns:**

None.

**Quality:**

3

**Strengths And Weaknesses:**

Strength:

1. The idea is novel. Although the representation, CodeQuery, is not proposed by the authors, using CodeQuery for LLM is reasonable as it makes the LLM easier to learn this format of code.

2. Introducing GRPO into the design of CAD is novel. Since RL is getting more and more attention in the research field of code generation of LLM, introduing into CAD design is reasonable and effective.

3. In line 265-280, the authors have demonstrates the details of implementation, which is very useful for reproduction.



Weakness:

1. The training curves of GRPO are not shown in the work. Showing how the reward and generation length is growing during training is extremely important in this work.

2. The authors does not show 3D meshes or rendered videos in supplementary results. I think using solely single-view images to demonstrate the results are not very good for demonstrate the full 3D results of this work.

3. Although some baselines are not open-sourced, currently only one baseline (Text2CAD) is compared in Table 1, which makes it not very solid.

---

> ### Author Rebuttal · Authors · 2025-07-31
>
> # Response to Reviewer 75fC
> Thank you for your constructive review. We appreciate your recognition of the novelty of using CadQuery as a structured intermediate representation for text-to-CAD generation, the introduction of GRPO into CAD modeling, and the reproducibility of our implementation details. These insights affirm the relevance and practicality of our proposed framework. We address your questions and concerns below, and will incorporate all suggestions into the next version of the paper.
> ## W1: GRPO Training Curve and Reward Progress.
> Due to rebuttal constraints, we are unable to include plots at this stage. Instead, we provide representative statistics demonstrating that the average reward increases while the generation length decreases during training, indicating effective policy learning. Although minor fluctuations are observed, the overall trend remains stable. We will include the full training curves and more detailed analyses in the next version.
> | train/global_step | RewardAPI | completion_length |
> |------------------:|----------:|------------------:|
> | **50**   | **0.41** | **538.98** |
> | 100   | 0.52 | 488.76 |
> | 150   | 0.43 | 524.22 |
> | 200   | 0.38 | 518.76 |
> | 250   | 0.38 | 540.30 |
> | **300**  | **0.50** | **480.01** |
> | 350   | 0.44 | 520.02 |
> | 400   | 0.44 | 510.65 |
> | 450   | 0.49 | 510.76 |
> | 500   | 0.57 | 504.23 |
> | **550**  | **0.60** | **515.47** |
> | 600   | 0.55 | 565.24 |
> | 650   | 0.61 | 508.30 |
> | 700   | 0.61 | 504.66 |
> | 750   | 0.55 | 522.70 |
> | 800   | 0.62 | 497.60 |
> | 850   | 0.66 | 497.33 |
> | 900   | 0.55 | 558.79 |
> | 950   | 0.58 | 525.95 |
> | 1000  | 0.63 | 503.71 |
> | **1050** | **0.67** | **505.29** |
> | 1100  | 0.62 | 517.74 |
> | 1150  | 0.60 | 496.27 |
>
> ## W2: 3D Mesh and Rendering Visualization.
> Due to format restrictions during the rebuttal phase, we cannot include 3D meshes or videos. We agree that single-view images are limited and will add multi-view renderings in the next version. Additionally, we will provide full meshes and interactive visualizations on the project page to better showcase qualitative results.
> ## W3: Limited Baseline Comparison in Table 1.
> Thank you for your question. Table 1 includes only one CAD-specific supervised baseline (Text2CAD[1]) because other recent methods like CAD-Translator[2] and CAD-LLaMA[3] are not open-sourced. Furthermore, their evaluation protocols (e.g., mesh normalization and preprocessing) lack sufficient detail for reproducible and fair comparison. To compensate, we include multiple LLM baselines under a unified prompt setting and provide qualitative visual comparisons.
> ## Q1: Failure Cases.
> As detailed in Sec. F of our Supplementary Material, common failure cases include: (1) misalignment between sub-components in complex assemblies, (2) confusion between operations such as extrusion and cutting, and (3) errors in thin structures or internal cavities due to sparse sampling. These issues highlight the challenges of fine-grained geometry modeling and reward design, which we aim to address in future work.
> ## Q2: Inference Time per Mesh Generation.
> We report the average time to generate a single mesh, including both reasoning (<think>) and code generation stages. The results below are measured across different GPUs:
> | GPU Model  | CoT Latency | SFT Latency |
> |------------|-------------|-------------|
> | H800 80G   | 0.06 s      | 0.03 s      |
> | A800 80G   | 0.18 s      | 0.12 s      |
> | 4090 24G   | 0.28 s      | 0.16 s      |
> | V100 32G   | 0.64 s      | 0.29 s      |
> ## Q3: GRPO Training Time
> As stated in Sec. A of our Supplementary Material, the GRPO stage was trained for 146 hours on 8×NVIDIA A800 (80GB) GPUs.
>
> [1] Text2cad: Generating sequential cad designs from beginner-to-expert level text prompts. Advances in Neural Information Processing Systems. NIPS 2024.
>
> [2] Cad translator: An effective drive for text to 3d parametric computer-aided design generative modeling. ACM MM 2024.
>
> [3] Cad-LLAMA: Leveraging large language models for computer-aided design parametric 3d model generation. CVPR 2025.

---

> > ### Author Response · Authors · 2025-08-05
> > **Any unclear explanations?**
> >
> > Dear Reviewers,
> >
> > Thanks for your efforts in reviewing this paper. We have tried our best to address the concerns and improve our work. Are there unclear explanations here? We can further clarify them.
> >
> > Best wishes,
> > Authors

---

### Official Review · Reviewer_rWzJ · 2025-07-01

**Clarity:** 3
**Significance:** 3
**Originality:** 2
**Rating:** 4
**Confidence:** 5

**Summary:**

This paper bridges text-to-CAD by introducing an intermediate representation in the form of CadQuery scripts and leveraging a large language model to generate CadQuery code from text descriptions. To implement this pipeline, the authors construct data from the Text2CAD dataset by feeding command sequences and text descriptions into a pretrained LLM to generate multiple CadQuery script candidates. These candidates are evaluated by computing the Chamfer Distance between the executed 3D models and the ground-truth models. The input description, the selected CadQuery script, and the corresponding CoT are then assembled into a corpus and used to fine-tune the pretrained LLM. The fine-tuning process follows the standard SFT+GRPO approach, where Chamfer Distance is used as the reward signal in GRPO. Experiments demonstrate the effectiveness of CAD-Coder compared to the original LLMs.

**Questions:**

1.	Max sequence length and compactness. Is there an issue with sequence length when using CadQuery scripts? For complex models, such as porous structures, how are long input and output sequences handled? What is the maximum sequence length supported by the LLM?

2.	Uncertainty in data construction. While I fully appreciate the training process (it's clearly explaine), the data construction raises some questions. If I understand correctly, the dataset contains only command sequences and text descriptions, not ground-truth CadQuery code. Instead, the CadQuery code is generated by the LLM and then filtered based on its distance from the ground-truth point cloud. My concern is, shouldn't there be a human review step to ensure the correctness of the scripts? If not, how can the accuracy of the generated CadQuery code be guaranteed?

3.	Why not generate CAD commands directly? As a follow-up to the previous question, why not directly generate the command sequence text instead of CadQuery code? Are there any experiments showing that using CadQuery code is superior to generating plain CAD command sequences?

4.	Natural language capabilities and conversational ability. Compared to prior work like Text2CAD, how well does this method preserve the LLM’s conversational and natural language understanding abilities?

**Ethical Concerns:**

["NO or VERY MINOR ethics concerns only"]

**Final Justification:**

I have carefully read the rebuttal and other discussions. I appreciate the rebuttal and decide to maintain the final rating.

**Limitations:**

As stated in the conclusion of the paper, the limitations include the lack of multi-modal input support, shallow CoT prompting, and the absence of fine-grained reward mechanisms.

From my perspective, I also question the model's interactive capabilities and its ability to perform fine-grained edits.

**Paper Formatting Concerns:**

No.

**Quality:**

3

**Strengths And Weaknesses:**

Strengths:

1.	A solid exploration of CAD code generation using large language models.

2.	The metrics demonstrate that the proposed method outperforms the original large language model and Text2CAD in terms of geometry quality and generation success rate.

3.	The paper is well-written and easy to follow.

Weaknesses:

1.	Technically sound but lacks novelty in the fine-tuning strategy. The paper directly adopts SFT+GRPO for refinement, so the main contribution lies in dataset construction rather than the training method. (However, this does not affect my overall rating.)

2.	The experiments do not clearly justify the need for intermediate CadQuery scripts.

3.	Based on the authors' description, there seems to be some uncertainty in the CadQuery script generation process. Using Chamfer Distance for filtering may also fail in certain cases. This concern is discussed in more detail under the Question section.

4.	Natural language capabilities and conversation: The paper lacks evaluation of the model’s inherent language understanding abilities.

---

> ### Author Rebuttal · Authors · 2025-07-31
>
> # Response to Reviewer rWzJ
> Thank you for your thoughtful and constructive review. We appreciate your recognition of our method as a solid exploration of CAD code generation with large language models, and your acknowledgement that our approach outperforms both the original LLM and Text2CAD[1] in geometry quality and success rate. These comments affirm the contribution of our intermediate CadQuery-based representation and training framework. We address your concerns in detail below and will incorporate your suggestions into the next version of the paper.
> ## W1: Novelty of the Fine-Tuning Strategy
> While our method does not introduce a novel reinforcement learning (RL) algorithm, we believe it makes a meaningful contribution by applying the pretraining + SFT + RL paradigm to structured code generation for 3D CAD models, a challenging task that requires both syntactic correctness and geometric fidelity. The design of our CoT prompts and reward functions is tailored to this task. To the best of our knowledge, this is the first work to validate RL with geometry-based rewards in the CAD domain. We hope this provides a strong foundation for future research on RL for CAD applications
> ## W2 & Q3: Justification for using CadQuery over Command Sequences.
> CadQuery offers distinct advantages as an intermediate representation. As described in Lines 35–41 of the Introduction, CadQuery provides richer semantics, a more compact structure, direct executability, and strong compatibility with Python-based LLMs. In contrast, traditional command sequences tend to be verbose, less abstract, and often require custom parsers. CadQuery supports modular design, reusable functions, and high-level geometric primitives (e.g., box, cylinder), making it more suitable for modeling complex structures and enabling downstream editing.
>
> Due to limited time and computational resources, we did not perform a direct comparison between generating CAD models as CadQuery scripts versus command sequences under identical experimental settings. As no prior works have conducted such a comparison, we agree that a controlled study under a unified setting would be a valuable direction for future research.
> ## W3 & Q2: Data Construction Uncertainty and Script Verification
> Thank you for your question. Your understanding is correct. Our CadQuery code is generated by the LLM and then filtered using Chamfer Distance (CD) and an executability check. We acknowledge that this automated pipeline may introduce some noise, as it does not involve comprehensive human review. However, manual evaluation is costly and subjective, since multiple scripts can yield the same geometry and script preferences may vary among annotators. CD offers an objective, geometry-based filtering criterion that scales efficiently and aligns with our downstream evaluation metrics.
>
> For constructing the CoT corpus, we do perform manual verification to ensure script correctness and clarity. In future work, we plan to explore human-in-the-loop or hybrid verification strategies to further improve data quality.
> ## W4 & Q4: Evaluation of Natural Language and Conversational Capabilities
> While our model is not explicitly trained on editing data, we observe that it can handle basic CAD editing tasks such as modifying dimensions, removing components, and adjusting positions (see Appendix E, Fig. S3), suggesting a degree of structural understanding and editing generalization.
>
> More broadly, our work focuses on improving geometric fidelity. Although we do not explicitly evaluate conversational or general-purpose language abilities, language understanding remains central to the task: the model must interpret prompts involving geometric intent, spatial constraints, and modeling actions, and generate structured CadQuery code. We build upon strong language models like Qwen2.5-7B-Instruct; while domain-specific fine-tuning may reduce generality to some extent, evaluating dialogue and fine-grained editing remains an open and underexplored direction across most prior works, including Text2CAD. We view this as an important opportunity for future research.
> ## Q1: Sequence Length and Handling of Complex Models
> Our model supports up to 131,072 input tokens and 8,192 output tokens, following the capacity of Qwen2.5-7B-Instruct. This is sufficient to handle complex CAD structures. This is a significant improvement over earlier methods such as Text2CAD (which supports up to 512 tokens), and allows for richer modeling capabilities. Thanks to CadQuery’s compact syntax, most CAD models (including those with multiple sketches) remain within 400–600 tokens. For highly complex cases (e.g., porous structures), we employ modular functions, parameterized loops, and hierarchical modeling to ensure the overall sequence length remains within the supported limit.
>
> [1] Text2cad: Generating sequential cad designs from beginner-to-expert level text prompts. NIPS 2024.

---

> ### Author Response · Authors · 2025-08-05
> **Any unclear explanations?**
>
> Dear Reviewers,
>
> Thanks for your efforts in reviewing this paper. We have tried our best to address the concerns and improve our work. Are there unclear explanations here? We can further clarify them.
>
> Best wishes,
>
> Authors

---

> > ### Comment · Reviewer_rWzJ · 2025-08-05
> >
> > The authors have addressed my technical concerns, and I believe that this is a novel pipeline. Although this is not going to affect my rating, I still think that the basic validation between is essential for an excellent paper (since the introduction is part of the novelty of this paper) and can not be simply ignored by "previous work does not conduct similar experiments".

---

> > > ### Author Response · Authors · 2025-08-06
> > >
> > > We sincerely thank the reviewer for taking the time to carefully review our work and engage with our rebuttal. Your constructive feedback in both the initial review and the follow-up comment has been extremely helpful in identifying valuable directions to strengthen our work. We truly appreciate your effort and thoughtful suggestions.
> > >
> > > We fully agree with your view: since the use of CadQuery as an intermediate representation is one of our core design decisions, a controlled comparison with command sequences would greatly strengthen the contribution.
> > >
> > > While such an experiment was not feasible within the timeline and scope of this submission, mainly due to parsing incompatibility and limited computational resources, we acknowledge that this validation is essential for a more complete justification of our representation choice.We have already planned this as a follow-up study and are actively preparing a unified evaluation pipeline that supports both command-sequence-based decoding and CadQuery generation under a consistent reward and execution framework.
> > >
> > > We appreciate your feedback once again and see this as an important opportunity to further improve the clarity and completeness of our work.

---

### Official Review · Reviewer_Sray · 2025-07-02

**Clarity:** 3
**Significance:** 4
**Originality:** 4
**Rating:** 5
**Confidence:** 4

**Summary:**

This paper introduces CAD-Coder, a new framework that translates natural language into CadQuery scripts. It features a two-stage training approach combining supervised learning and reinforcement learning with GRPO, along with a chain-of-thought planning process. The results show that CAD-Coder improves the ability of LLMs to generate accurate, complex CAD models from text.

**Questions:**

There are some experimental details that need to be confirmed.
1. How does the method adapt to multi-component, highly-constrained assemblies (including complex features like rotation, lofting, chamfering, etc.)? What are the anticipated key technical challenges?
2. After generating \<think\> reasoning, how much does the average latency per sample increase? In editing scenarios, how does time consumption scale with script length?

**Ethical Concerns:**

["NO or VERY MINOR ethics concerns only"]

**Final Justification:**

The authors have adequately addressed most of the points raised regarding the evaluation metrics and the comparison with SOTA methods. The reviewer has no further questions.

**Limitations:**

yes

**Quality:**

3

**Strengths And Weaknesses:**

Strengths:
1. The paper is well-written and logically structured, with clear explanations.
2. The proposed method is straightforward and technically sound, with accessible implementation details.
3. The geometry reward in GRPO training appears to be a notable and reasonable contribution, as it shifts from language-level loss optimization to direct end-to-end optimization of geometric quality.
4. I think Converting CAD command sequences into CadQuery code is a practical and intuitive approach, enabling direct manipulation of geometric parameters through code.

Weaknesses:

1. Although topological rewards are introduced, quantitative results still do not adequately reflect multi-dimensional indicators, making it difficult to comprehensively assess industrial applicability.
2. Lacks direct comparison with recent CAD-specific generators (such as CAD-Translator, CAD-LLaMA), somewhat weakening experimental persuasiveness.
3. RL fine-tuning requires 8×A800 GPUs and 146 hours. The paper does not yet discuss lightweight approaches or migration strategies for small-to-medium models, or impacts on actual deployment.

[1] Cad translator: An effective drive for text to 3d parametric computer-aided design generative modeling. ACM MM 2024.
[2] Cad-llama: Leveraging large language models for computer-aided design parametric 3d model generation. CVPR 2025.

---

> ### Author Rebuttal · Authors · 2025-07-31
>
> # Response to Reviewer Sray
> Thank you for your constructive and encouraging review. We appreciate your recognition of our method’s technical soundness, the practicality of translating CAD command sequences into CadQuery scripts, and the value of introducing geometry-aware rewards in GRPO for direct optimization. These insights affirm the relevance of our contributions to the CAD and LLM community. We respond to your comments below and will incorporate your suggestions into the next revision.
> ## W1: Evaluation Metrics
> Our choice of the Chamfer Distance (CD) metric is motivated by the core requirements of the Text-to-CAD task, where the primary goal is achieving sample-level geometric fidelity. While metrics such as MMD or JSD are useful for assessing diversity, they do not capture the accuracy of individual CAD models. In contrast, CD directly quantifies the geometric error between the generated shape and the ground truth, which is crucial for the high-precision demands of CAD applications. Adopting CD also ensures fairness and comparability with existing methods, such as Text2CAD[1], which also use CD as the primary metric.
> ## W2: Lack of Comparison with CAD-Translator and CAD-LLaMA
> A direct comparison with CAD-Translator [2] and CAD-LLaMA [3] is currently infeasible as their models and code are not publicly available. In addition, their mesh normalization and preprocessing strategies (e.g., scaling and alignment) differ significantly, resulting in Chamfer Distance values that differ by an order of magnitude from ours. As a result, we are unable to fairly or reproducibly compare against the metrics reported in their papers. To ensure a rigorous evaluation, we instead adopt the established Text2CAD pipeline to compute Chamfer Distance (CD) for our method and all baselines.
> ## W3: Lightweight Strategies and Model Portability
> While our full-scale RL tuning utilizes 8×A800 GPUs, the framework (LLaMA-Factory + VERL) supports flexible scaling.
>
> **Lightweight Strategies**: Smaller models (1B–3B) can be trained on a single GPU in 20–40 hours.
>
> **Model Portability**: At deployment, we support 4-bit/8-bit quantization and vLLM-based acceleration. Inference latency remains within seconds, enabling real-time usage. Our current setup serves as full-scale validation; lightweight adaptation is fully supported.
>
> We also provide inference latency measurements across different GPU types (per sample):
> | GPU Model  | CoT Latency | SFT Latency |
> |------------|-------------|-------------|
> | H800 80G   | 0.06 s      | 0.03 s      |
> | A800 80G   | 0.18 s      | 0.12 s      |
> | 4090 24G   | 0.28 s      | 0.16 s      |
> | V100 32G   | 0.64 s      | 0.29 s      |
> ## Q1: Adaptation to Assemblies
> Thank you for your question! Our framework is well-suited for modeling complex assemblies. CadQuery supports advanced features (e.g., rotation, lofting, chamfering), and our Chain-of-Thought planning naturally enables hierarchical modeling of components and their spatial relationships. The primary challenges are data-related: existing datasets predominantly contain single-component objects and lack annotated assemblies. Additionally, assemblies involve strong constraints (e.g., tolerances, fits) that are not captured by geometry-only metrics such as CD. In summary, while our modeling architecture is extensible to assemblies, scaling to realistic assembly tasks will require richer datasets and constraint-aware learning.
> ## Q2:  Latency and Editing-Time Scaling
> The latency impact of the `<think>` reasoning step is already addressed in W3, where we provide detailed measurements across different hardware setups.
>
> Editing capability is an early-stage exploration of our model's potential and is not a primary focus of this work. We have not yet conducted a systematic evaluation of how inference time scales with script length in editing scenarios. However, because most edits involve localized changes, input/output sequences are typically much shorter than for full-script generation. Therefore, the end-to-end generation latencies reported above can be viewed as a conservative upper bound for editing tasks.
>
> [1] Text2cad: Generating sequential cad designs from beginner-to-expert level text prompts. NIPS 2024.
>
> [2] Cad translator: An effective drive for text to 3d parametric computer-aided design generative modeling. ACM MM 2024.
>
> [3] Cad-LLAMA: Leveraging large language models for computer-aided design parametric 3d model generation. CVPR 2025.

---

### Official Review · Reviewer_vKK8 · 2025-07-02

**Clarity:** 3
**Significance:** 2
**Originality:** 3
**Rating:** 4
**Confidence:** 4

**Summary:**

This paper presents CAD-Coder, a text-conditioned large language model (LLM) for generating CAD models in the form of CadQuery code. The proposed pipeline first undergoes supervised fine-tuning, followed by Group Reward Policy Optimization using a geometry-aware reward based on Chamfer Distance and an auxiliary format-consistency reward. To support training, the authors introduce a dataset of 110k triplets created by translating Text2CAD command sequences into CadQuery code. CAD-Coder achieves lower Chamfer Distance compared to Text2CAD and general-purpose LLMs.

**Questions:**

- Could the authors address the concerns on the data synthesis pipeline (first mention weakness above)?

- Could the authors provide details on how the GPT-4o baseline was constructed, including prompt design and use of examples? What explains the relatively low 93% instruction rate, given recent improvements in LLMs?

- Could the authors elaborate on the training setup for the SFT baseline and explain its underperformance, especially in simple cases? Is this potentially linked to limitations in the data synthesis process?

**Ethical Concerns:**

["NO or VERY MINOR ethics concerns only"]

**Final Justification:**

The authors addressed most of the concerns raised in the rebuttal (SFT performance and data quality). The reviewer recommends that the authors incorporate the clarifications and the discussions in the updated version of the paper.

**Limitations:**

The limitations are addressed in a dedicated paragraph.

**Paper Formatting Concerns:**

No major formatting issues identified by the reviewer.

**Quality:**

3

**Strengths And Weaknesses:**

**Strengths**

- Text to CAD is rapidly gaining attention and the community needs stronger baselines on popular LLMs. The reviewer thinks that this paper is well aligned with current interests.

- Switching from CAD token sequences to a Pythonic CAD API such as CADquery brings multiple benefits. CADquery is well suited for generation via LLMs, compact and editable by engineers.

- The authors adapt proven LLM techniques to CADQuery generation, showing how methods like SFT, CoT, and GRPO can be effectively transferred to structured code tasks, offering useful insights for both CAD and ML research.


**Weaknesses**

-  The data synthesis pipeline appears suboptimal. Intuitively, this translation task is not especially difficult, given that the LLM is provided with the full input sequence. However, the quality of the translated data significantly impacts the reported performance throughout the paper. The proposed translation method yields near-identical results for only a small subset of the samples (8k out of 178k with very low Chamfer Distance). The authors themselves note that training on a larger dataset of medium-quality translations leads to worse performance. Moreover, the test set is generated using the same translation pipeline, which fails for a large portion of the data (32k out of 178k, or about 17%). This raises concerns about the fairness of comparisons with Text2CAD, given that the evaluation relies on noisy test data.

-  The quality of the baseline comparisons raises concerns. In the reviewer's experience, GPT models are increasingly capable of generating valid CadQuery code. However, in Table 1, the authors report only a 93% instruction rate (IR) for GPT-4o, which suggests suboptimal prompting, potentially due to inadequate instructions or a lack of in-context examples. Moreover, the paper provides no details on how the LLM baselines were constructed, making it difficult to assess the validity of the comparison.

- The reviewer is also concerned about the results in Table 2, particularly the poor performance of the supervised fine-tuning (SFT) baseline. It is surprising that SFT underperforms, especially considering that Text2CAD, which uses a significantly smaller model than Qwen2.5-7B-Instruct, achieves better results without any reinforcement learning. This discrepancy is also visible in Figure 3, where the SFT variant fails to generate simple geometries in response to straightforward prompts, such as those shown in the first row. These unintuitive results may be linked to the limitations of the data synthesis pipeline.

- Given the paper's focus on Pythonic CAD generation, the proposed data synthesis approach could be extended to other aspects of code quality, such as adding comments, choosing meaningful variable names, or structuring code more clearly.

- CADAssistant is incorrectly described as using CadQuery, whereas it actually relies on the FreeCAD Python API.

- Authors do not clarify if they will release the constructed CADQuery dataset

---

> ### Author Rebuttal · Authors · 2025-07-31
>
> # Response to Reviewer vKK8
> Thank you for your thorough review and constructive feedback. We appreciate your recognition of our work’s alignment with current research trends, the benefits of using CadQuery as a Pythonic CAD API, and the effective adaptation of LLM techniques (SFT, CoT, GRPO) to structured CAD code generation. These comments are encouraging and helpful. We address your concerns in detail below and will incorporate all suggestions in the next version of the paper.
> ## W1 & Q1: Data Synthesis Pipeline
> **Challenge of Identical Model Generation**: Obtaining identical models by translating from command sequences to CadQuery scripts is not straightforward. The mapping between CAD command sequences and CadQuery scripts is inherently one-to-many: a given command sequence can be implemented through various syntactically different, yet functionally equivalent, CadQuery scripts. Consequently, translated CadQuery scripts may differ from the original command sequences, leading to potential discrepancies.
>
> **Data Quality**: It is important to clarify that the 8k samples we use are the highest-fidelity subset. We strategically selected this subset for the initial Supervised Fine-Tuning (SFT) stage to ensure the model acquires a robust syntactic and semantic foundation. Our experimental results (Table 3) validate our choice, and we believe this insight is valuable for future work, i.e.,  for SFT, especially when followed by post-training, data quality overweights quantity.
>
> **Test Set Fairness**: We apologize for any confusion. The evaluation set remains identical to that of the Text2CAD[1] dataset to ensure fair comparison. Specifically, during evaluation, we compute the Chamfer Distance between the generated models (from predicted CadQuery scripts) and the ground truth CAD models provided by DeepCAD. Thus, the comparison across methods is fair and consistent.
> ## W2 & Q2: Baseline Comparison Quality
> **Baseline Details**: To ensure a faire comparison, we use the same CoT prompt for all LLM baselines, as described in Sec. C of the Supplementary Material.
>
> **GPT-4o Performance**: Regarding GPT-4o’s performance, we attribute its results to several factors. (1) The primary challenge is the domain-specific complexity of our data, which consists of expert-level prompts that general-purpose models often struggle to interpret. (2) We also observed that deprecated CadQuery syntax (due to API updates) may negatively affect GPT-4o’s outputs. To address this, we re-evaluated GPT-4o by accepting generated CadQuery code as valid if it is functionally correct, even when using deprecated syntax. Under this criterion, its invalid rate (IR) drops to 70.5%. Notably, a similar invalid rate (74.26%) was reported for GPT-4o in the recent CADFusion[2] paper, indicating a broader limitation of general-purpose LLMs on CAD-specific generation tasks. Furthermore, following your suggestion, we experimented with in-context learning by providing an example to GPT-4o; however, this resulted in even lower performance.
> ## W3 & Q3: SFT Performance
> Thank you for raising this point. The underperformance of our SFT-only model is expected, as it serves merely as a lightweight initialization for our core GRPO stage, not as the final model. We deliberately train SFT on a small, high-purity dataset (8k samples) to teach basic CadQuery syntax, providing a solid foundation for subsequent reinforcement learning. Therefore, comparing the SFT-only model’s performance to the fully supervised Text2CAD model (trained on 178k samples) is not a fair comparison.
> As shown in Fig. 3, most examples represent complex cases, demonstrating the superiority of our final model after incorporating the GRPO stage. Fig. 5(a) indicates that the SFT-only model performs well on simpler shapes, with over 4,500 samples achieving a Chamfer Distance below 1e-3, confirming that it learns a solid foundational understanding.
> ## W4: Code Quality Extensions
> Thank you for your suggestion. As shown in Sec. C of the Supplementary Material, our CoT prompts encourage code comments during data synthesis, which we found improves the model’s understanding of geometric intent. We agree that using meaningful variable names and clearer code structuring would further enhance data quality, and we plan to incorporate these improvements in future work.
> ## W5: CADAssistant Clarification
> Thank you for pointing this out. We will correct it in the next version.
> ## W6: Dataset and Code Release
> We are committed to fully open-sourcing the following upon paper acceptance: the constructed CADQuery dataset, model weights, and inference code.
>
> [1] Text2cad: Generating sequential cad designs from beginner-to-expert level text prompts. NIPS 2024.
>
> [2] Text-to-CAD Generation Through Infusing Visual Feedback in Large Language Models. ICML 2025.

---

> > ### Comment · Reviewer_vKK8 · 2025-08-04
> >
> > The reviewer thanks the authors for their answers. The rebuttal addressed most of the concerns, the reviewer does not have any other questions.

---

> > > ### Author Response · Authors · 2025-08-06
> > >
> > > We sincerely thank Reviewer vKK8 for their time, thoughtful review, and constructive suggestions. We appreciate your engagement with our rebuttal and are glad that our responses addressed your concerns. Your comments were instrumental in helping us refine our explanation of the data synthesis pipeline, baseline evaluation, and fine-tuning strategies. We will incorporate all your feedback in the next version and look forward to improving the clarity and rigor of this work further.

---

### Decision · Program_Chairs · 2025-09-17

**Decision:**

Accept (poster)

**Comment:**

This paper presents CAD-Coder, a framework for text-to-CAD generation that reformulates the task as CadQuery script generation, supported by supervised fine-tuning, reinforcement learning with geometry-aware rewards, and chain-of-thought reasoning. The authors also contribute a large-scale dataset and show strong experimental results, outperforming Text2CAD and general-purpose LLMs on geometry fidelity and validity. While reviewers noted some limitations in data construction, baseline coverage, and evaluation breadth, the rebuttal provided convincing clarifications and additional evidence on fairness, data quality, efficiency, and generalization. The contributions are timely, technically solid, and well aligned with growing interest in text-to-3D and structured code generation. Overall, this is a clear accept.